# Cervical Myelopathy in Patients Suffering from Rheumatoid Arthritis—A Case Series of 9 Patients and A Review of the Literature

**DOI:** 10.3390/jcm9030811

**Published:** 2020-03-17

**Authors:** Insa Janssen, Aria Nouri, Enrico Tessitore, Bernhard Meyer

**Affiliations:** 1Department of Neurosurgery, University of Geneva, 1205 Geneva, Switzerland; arianouri9@gmail.com (A.N.); Enrico.Tessitore@hcuge.ch (E.T.); 2Department of Neurosurgery, Klinikum rechts der Isar, Technical University Munich, 81675 Munich, Germany

**Keywords:** cervical myelopathy, spinal cord compression, cervical spine surgery, rheumatoid arthritis (RA), cranial settling (CS), atlantoaxial subluxation (AAS), atlantoaxial instability (AAI)

## Abstract

Cervical myelopathy occurs in approximately 2.5% of patients suffering from rheumatoid arthritis (RA) and is associated with notable morbidity and mortality. However, the surgical management of patients affected by cervical involvement in the setting of RA remains challenging and not well studied. To address this, we conducted a retrospective analysis of our clinical database between May 2007 and April 2017, and report on nine patients suffering from cervical myelopathy due to RA. We included patients treated surgically for cervical myelopathy on the basis of diagnosed RA. Clinical findings, treatment and outcome were assessed and reported. In addition, we conducted a narrative review of the literature. Four patients were male. Mean age was 64.8 ± 20.5 years. Underlying cervical pathology was anterior atlantoaxial instability (AAI) associated with retrodental pannus in four cases, anterior atlantoaxial subluxation (AAS) in two cases and basilar invagination in three cases. All patients received surgical treatment via posterior fixation, and in addition two of these cases were combined with a transnasal approach. Preoperative modified Japanese orthopaedic association scale (mJOA) improved from 12 ± 2.4 to 14.6 ± 1.89 at a mean follow-up at 18.8 ± 23.3 months (range 3–60 months) in five patients. In four patients, no follow up was available, and the mJOA of these patients at time of discharge was stable compared to the preoperative score. One patient died two days after surgery, where a pulmonary embolism was assumed to be the cause of mortality, and one patient sustained a temporary worsening of his neurological deficit postoperatively. Surgery is generally an effective treatment method in patients with inflammatory arthropathies of the cervical spine. Given the nature of the RA and potential instability, fixation in addition to cord decompression is generally required.

## 1. Introduction

Cervical myelopathy occurs in approximately 2.5% of patients suffering from rheumatoid arthritis (RA) for more than 14 years and is associated with notable morbidity and mortality [1]. In these cases, compression of the spinal cord and brain stem is typically caused by atlantoaxial instability (AAI) or atlantoaxial subluxation (AAS) with associated vertical migration of the dens of C2 and accompanying retrodental pannus formation [2].

Rheumatoid arthritis is a chronic inflammatory systemic disease and affects about 1–2% of adults [3]. Besides peripheral joints, the upper cervical spine is preferentially afflicted in patients with RA due to the anatomic vulnerably of numerous synovial and apophyseal joints to dynamic forces [4,5]. This is especially true for the cranio-cervical junction and the atlantoaxial joints. In these patients, an inflammatory synovial proliferation results in deterioration of ligamentous tendinous tissues and bone erosion, thereby increasing sliding motion between the atlas and axis, and as a consequence, atlantoaxial instability (AAI) [5]. In the advanced stage, an atlantoaxial subluxation (AAS) can develop and manifest in the vertical migration of the dens, known as cranial settling (CS), basilar impression, basilar invagination or vertical translocation (VT) [4,6]. Most common is an anterior atlantoaxial subluxation due to an affected anterior median atlanto-axial joint between ventral arch of C1 and the dens of the axis (circa 75% of all AAS). Subluxation in the posterior median atlanto-axial joint, located between posterior arch of C1 and the dens of the axis, as well as asymmetrical or unilateral changes of the lateral atlanto-axial joint leading to impairment in rotation are rare (7% and 20%) [7]. For diagnosis of cervical involvement in RA various diagnostic imaging options are available, each playing a different role depending on the stage of involvement (see discussion, Figure 1 and Figure 2, Table 1).

After the introduction of early and consistent therapy with biologicals, the prevalence of cervical involvement in RA has been declining and is estimated to be between 20–44%, whereas prevalence rates in historical data present estimates between 42–80% [9]. Symptomatic involvement of the cervical spine in terms of instability manifests itself on average about 7–10 years after initial diagnosis of RA [3,10]. Atlantoaxial subluxation with myelopathy is seen in approximately 2.5% of patients with RA for more than 14 years [5]. For prevalence and symptoms, see Table 2. Medical therapy of RA at the early stages is feasible to prevent cervical involvement in RA. Early diagnosis and thus an early start of a sufficient drug therapy is regarded as a prognostically positive factor with regard to a risk reduction for a manifestation of cervical involvement [11]. Thanks to drug therapies, the instabilities requiring surgery are statistically decreasing, however, the cases needing treatment are becoming more complex [12].

While conservative therapy can effectively palliate pain, operative treatment is indicated in medically therapy resistant pain with radiographic instability, and is necessary after the development of neurological deficit, myelopathy, cranial nerve and bulbar dysfunction [8].

The objective of the present paper is to display the variety of disorders that present and discuss this in the management of such cases in the context of current literature.

## 2. Methods

A retrospective analysis of the clinical database of the department of neurosurgery in Munich was performed between May 2007 and April 2017, with research ethics board approval (Ethics committee of Technical University of Munich, No.238/17 S).

We searched the database for patients which had received surgery for craniocervical and subaxial instabilities. Patients with traumatic or neoplastic retro-odontoid pannus or craniocervical junction abnormalities like chiari malformation were not included in this series. Out of this cohort, we investigated patients treated surgically for cervical myelopathy on the basis of diagnosed or suspected rheumatoid arthritis or chronic inflammatory systemic disease. We excluded patients with pure degenerative instabilities. Clinical findings, treatment and outcome were assessed. In addition, we conducted a review of the literature. The aim of the study was to evaluate variations in patient characteristics, surgical treatment and patient outcomes. To assess the cervical involvement and severity of myelopathy, the Ranawat classification (Table 3) and criteria and the modified Japanese orthopaedic association scale (mJOA); modified by Keller, 1993) were used to assess degree of neurological function [7,14]. Post-operative mJOA was obtained at the last follow-up, at least 3 months following operation.

## 3. Results

In the defined period of time, 39 patients received surgery for rheumatic and degenerative cervical instabilities. Six patients were known for RA and under medical treatment. In five patients a chronic inflammatory systemic disease was suspected but not diagnosed at time of neurosurgical treatment. Out of 39 patients, myelopathy was present in 16 patients (47.1%).

Out of eleven patients known for rheumatoid arthritis (RA) or suspected chronic inflammatory systemic disease, 9 patients were suffering from myelopathy. Four patients were male, five females. Mean age was 64.8 ± 20.5 years (range 22–82 years). All patients presented with clinical myelopathy, and additionally some patients presented with further neurological deficits (see Table 4 and Table 5). Surgeries were performed by seven surgeons. The underlying cervical pathology was AAI of the median atlanto-axial joint combined with retrodental pannus in four, anterior AAS in two and basilar invagination in three cases. Four patients were already known for RA and under medical treatment. In five patients a chronic inflammatory systemic disease was suspected but not diagnosed at time of neurosurgical treatment. The diagnosis for RA was confirmed later by a rheumatologist in each case. Patients were classified according to the Ranawat criteria (see Table 3 and Table 5).

Mean preoperative modified Japanese orthopaedic association scale (mJOA) at admission was 12.67 ± 2.83 (range 8-16). All patients were suffering from symptoms for several month and sustaining a progressive worsening within few weeks before surgical treatment. Five patients were observed with a mean follow-up at 18.8 ± 23.3 months (range 3–60 months), all of which showed improvement of pain and neurological deficits. The mean preoperative mJOA score in these patients showed an improvement from 12 ± 2.4 to 14.6 ± 1.89, but only one patient made a complete recovery (case 2). In four patients, no follow up was available, and the mJOA of these patients at time of discharge was stable compared to the preoperative score.

Surgical treatment via posterior fixation was conducted in all cases. In the presence of AAI (n = 2) and AAS (*n* = 2) associated with retrodental pannus, patients underwent a posterior C1-C2 stabilization using the Goel-Harm’s technique. Due to an associated subaxial spinal stenosis two patients underwent a cervical fixation from C1 to C3 and C1 to C6. In patients of underlying basilar invagination, stabilization including C0 was performed (*n* = 3). Posterior decompression was necessary in six cases.

In two cases, a ventral decompression in terms of a transnasal endoscopic dens resection was performed, whereas in one case the operation took place two years after the first intervention (case 9). A 50- year old patient first showed an improvement of myelopathy after posterior fixation C0-2 and suboccipital de-compression for basilar invagination, but she again sustained a worsening of symptoms after two years. The other patient was diagnosed for AAI with retrodental pannus.

One patient suffering from basilar invagination resulting in tetraparesis (Ranawat class IIIB) died on the second postoperative day. The cause of death remained unclear, but was suspected to be due a pulmonary embolism. In one patient (Ranawat class IIIA) a temporary worsening of the neurological deficit after posterior fixation for AAI and retrodental pannus occurred. He received a second surgery for transnasal ventral decompression and mainly recovered from his hemiparesis. Among the other patients, there was no complication, and none showed a new neurological deficit (Table 4 and Table 5).

### Illustrative Case (2)

A 22-year-old female patient presented at emergency with rapidly progressive fine motor disorders and paresis of the right arm. On clinical examination, mild urinary retention as well as ataxia was also observed, and she scored 12 on the mJOA scale. The patient was known for RA for several years and was treated with Adalimumab every two weeks. Imaging showed an atlantodental instability with retrodental pannus formation and basilar invagination. Basilar invagination was confirmed by a Ranawat criteria of 10.7 mm and an anterior atlantodental interval of 9.7 mm, indicating a Ranawat IIIA classification. The first step was a closed reduction under X-ray control and the application of a rigid neck brace on the day of admission in order to treat the symptoms immediately. In the second step, open reduction and fixation via implantation of an internal fixator took place two days later.

The nerve root C2 was exposed and rhizotomized on both sides, the facet joint between C1/2 distracted and autologous bone chips inserted. After three months, residual sensory deficit of the fingertips on both sides remained, a motor deficit, urinary retention and ataxia were no longer present. There was no relevant pain symptomatology. After 12 months, she had made a complete recovery, and scored 17 on the mJOA scale—see Figure 3.

## 4. Discussion

Our results show that surgical treatment is a very effective treatment method in patients with inflammatory arthropathies of the cervical spine. The patients available for follow up showed an improvement of preoperative mJOA from 12 ± 2.4 to 14.6 ± 1.89, the others showed a stable score at time of discharge without a new neurological deficit. However, in our case series two patients sustained a relevant complication, with one patient dying the second postoperative day (likely due to a pulmonary embolism) and the other experiencing a temporary worsening of a hemiparesis after posterior fixation for AAI and retrodental pannus was achieved. The latter patient received a second surgery via transnasal ventral decompression and mainly recovered from his hemiparesis in the following months. Both of these patients presented with a severe neurological deficit (preoperative mJOA 8 and 9, Ranawat class IIIA and B) which confirms the assumption that perioperative morbidity and mortality increases significantly in rheumatoid non ambulatory (Ranawat Class IIIB) patients with loss of the ability to walk [8,16].

In case of isolated AAI without subluxation C1-C2 fixation associated with (*n* = 5) and without (*n* = 4) C1 laminectomy was performed. In one case disappearance of the pseudotumor failed, and a ventral approach was necessary after two years, which was conducted endoscopically via a transnasal approach. 

Pain in the craniocervical junction is a typical symptom that is seen in 69% of patients with cervical instability [3]. Occipital neuralgia, facial hypoesthesia, facial pain as well as pain in the mastoid area and ear pain may also manifest as they are caused by irritation of the occipital nerve between C1 and C2 or by irritation of the trigeminal nucleus in the medulla oblongata [3]. Patients can also suffer from radiculopathy of the upper extremities. Neural deficits like myelopathy, muscular atrophy, paresis, bladder rectal disorders, pathological reflexes and spasticity are present in up to 58% of all cases [6,17,18]. Compression of the medulla oblongata can also lead to disorders of the caudal cranial nerves such as dysphagia and dysarthria resulting from compression of the vagus, glossopharyngeal and hypoglossal nerve. Involvement of the cranial nerves is reported in about 20% of cases [3]. Beyond the development of severe neurological symptoms, locked-in syndrome or sudden death can also occur [17,19]. Mechanical compression can cause the development of a vertebrobasilar insufficiency with tinnitus and dizziness. Similarly, instability can cause a kinking of vertebral arteries, which can lead to vertebrobasilar thromboembolic events. Another, but less common consequence of RA, is the destruction and instability of the subaxial spine resulting in subaxial dislocation (SAS) [3,6,17]. Aseptic discitis and atraumatic dens fractures have also been reported and can also be the consequence of the RA inflammatory process [3]. For prevalence and symptoms, see also Table 1.

Traditional imaging is conducted using conventional x-ray of the cervical spine, which gives information about spinal alignment. As it is related to lower cost, it is widely available and has a low radiation dose, it is especially suitable for screening of asymptomatic patients with RA in the outpatient setting [5,6]. Additionally, flexion- extension images allow for easy visualization of occult instabilities [6]. However, the gold standard for bone evaluation and evaluation of ankylosis and pseudarthrosis is the cervical CT scan, which is also often acquired for surgical planning. For the assessment of soft tissue, spinal cord, or nerve compression and myelopathy, cervical MRI is the imaging modality of choice. It is always indicated for the evaluation of patients with neurology deficit [6].

The conventional x-ray of the cervical spine is conducted in anterior-posterior and lateral projection, supplemented by flexion-extension images as well as radiographs through the open mouth aimed at the odontoid process [5,6]. In case of abnormalities in conventional images, clinical suspicion or neural deficits, cervical CT and MRI are added. By definition, an atlantoaxial subluxation is present if the atlantodental interval (AADI) measured on lateral radiographs is >3 mm. Evidence of AAS is also shown by a distance between the dens and C1 posterior arch (posterior atlantodental interval (PADI) <14 mm [7].

For the assessment of vertical subluxation (VS) a number of methods have been developed to describe the degree of dens displacement with regard to foramen magnum. One popular diagnostic method is the Ranawat criteria, which is based on a line which connects the center of the anterior arch with the center of the posterior arch of C1 vertebra and a second line which is drawn along the axis of the odontoid process, from the center of the base of C2 vertebra to the intersection with the first line (Figure 1). The distance is used to assess the presence of the basilar invagination and based on the criterion are defined to be present when < 13 mm in women and < 15 mm in men [7]. Other diagnostic criteria of cranial settling are described in Figure 2. SAS is diagnosed when the radiograph in lateral projection shows horizontal displacement of vertebrae with irreducible translation by >3.5 mm (Table 1) [7].

Besides the grade of AAS, the localization and the extend of neurological impairment, atlanto-axial instability can be classified as reducible, partially reducible or fixed, according to the response to traction [4]. RA with cervical involvement is a progressive and serious condition with reduced lifetime expectancy. Matsunaga et al. noted that all of the irreducible AAS patients who did not undergo surgical treatment were bedridden within 3 years after the onset of myelopathy, and the survival rate was 0% in the first eight years [15,20]. Atlantodental involvement is typically a late manifestation of RA [3]. However, patients with a worse disease activity score, high disease activity and erosive disease at baseline have a high risk of atlantoaxial involvement in early rheumatoid arthritis (ERA) (disease duration < 12 months) [1]. Affection of the small joints of the hand and foot, failure of antirheumatic therapies, intake of glucocorticoids, young age at diagnosis, level of CRP, female sex, and low BMI were identified as risk factors for cervical involvement [21,22].

In the early stage, clinical involvement is initially asymptomatic in 33–50% of patients, which makes early drug intervention before the development of instability more difficult. Symptoms of cervical myelopathy are often misinterpreted due to the rheumatic affection of the finger and ankle joints, which may mask myelopathy as the cause of fine motor disorders and gait insecurity.

In case of isolated AAS without vertical instability or relevant destruction of the C0-C1 joints, atlantoaxial fusion via C1-C2 fixation is the means of choice [17]. The stabilization should only be extended to C0 if not otherwise possible, as this means an important limitation of flexion and extension [23]. The atlanto-axial transarticular screw fixation introduced by Magerl has been the gold standard for treatment of atlanto- axial instability for long time [24]. At present, due to reduced risk for neurovascular complications, the surgical technique of choice is the technique described by Harms and Goel, containing C1 lateral mass and C2 pedicle screws which has proven its efficacy and safety [25]. In comparison to transarticular screw fixation, the Harms technique also includes reduction of C1-C2 and the protection of the C1-C2 joint [15].

Direct removal of intracanalar tissue compressing the spinal cord is not obligatory [17]. After fixation of an instability by posterior stabilization, the retrodental pannus usually recedes [2].

Anterior decompression may be indicated for cases that are irreducible or in cases in which a pannus fails to regress after posterior fixation. By distraction of facet C1-C2 an improvement of the cranial setting is possible [17].

The transoral dens resection is known as the classical method for a ventral decompression, but it is associated with considerable risks. In 33.6% of cases the splitting of the soft palate is necessary, resulting complications include hypernasal speech, nasal regurgitation, tracheal edema and fistula formation in the pharynx. As a result, 14.8% of patients require a PEG and 3.8% a tracheotomy. A velopharyngeal insufficiency results in 4% of cases, and the mortality rate is 2.3% [26,27]. In a case series by Gempt et al. the transnasal endoscopic approach could be shown as an alternative to transoral decompression [28]. If a ventral dens resection is necessary, the endoscopic transnasal approach is preferred. Compared to the transoral approach, the patients can be extubated after 0.3 days instead of 3.5 days after transnasal endoscopic access, and oral nutrition can be started after one day instead of 5.2 days [26,27].

Given the fact that a relevant number of patients are asymptomatic in the stage of cervical involvement, the objective beside the optimal surgical management should be the early detection of cervical involvement. In a study by Neva et al., a previously undiagnosed cervical subluxation could be detected in 44% of a total of 194 patients included who underwent an orthopaedic surgery of the peripheral joints due to RA (Mean age was 65 years, mean duration of RA was 24 years). Compared to those without subluxation, the affected patients did not show a more frequent presence of symptoms such as neck pain, occipital, temporal, retro-orbital pain or radiculopathy of the upper extremities. Thus, by clinical appearance alone, patients with subluxation are not distinguishable from those without [29]. Therefore, a radiological screening of patients with RA should be considered.

## 5. Conclusions

Surgical treatment is an effective treatment method in patients with inflammatory arthropathies of the cervical spine, whereas the risk of perioperative morbidity and mortality increases significantly with the severity of neurological impairment. Due to the advent of disease modifying medication, the prevalence of cervical involvement in RA seems to be diminishing. However, when present, these cases can be challenging to treat. Given the dynamic nature of RA myelopathy, surgical treatment generally requires fixation and was performed in all patients in the presented cohort.

## Figures and Tables

**Figure 1 jcm-09-00811-f001:**
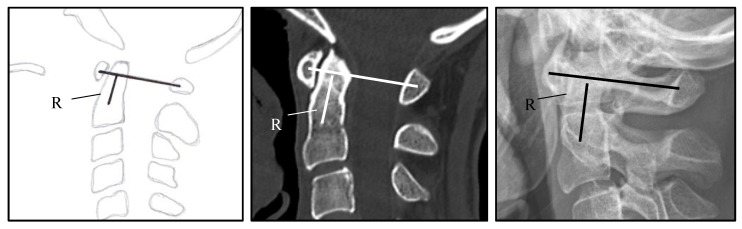
Ranawat criteria: Line from the center of the anterior C1 arch to the center of the posterior C1 arch. A second line goes through the axis from the odontoid to the center of the base of the C2. The smaller the distance (R) to the crossing of the lines, the more pronounced the invagination. A distance > 13 mm for women and > 15 mm for men is normal [8].

**Figure 2 jcm-09-00811-f002:**
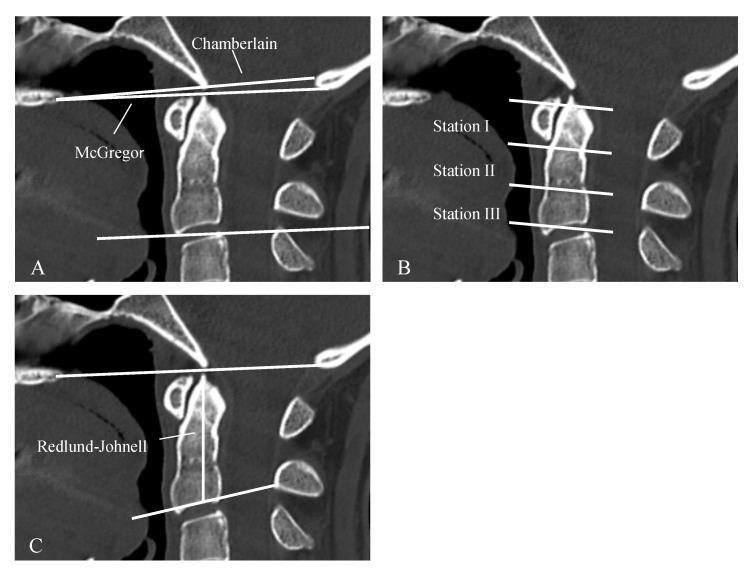
Various diagnostic criteria of cranial settling. Chamberlaine: Line from the dorsal end of the hard palate to the posterior border of the foramen magnum. If the dens is 3 mm above the line, a basilar invagination is present by definition (**A**). McGregor: Line from the hard palate to the deepest point of the occiput. A basilar invagination is present if the tip of the dens is more than 4.5 mm above the line (**A**). Clark-Station: Here the dens is divided into three equally sized stations. If the front arch of the atlas reaches into station II or III the criteria of a cranial settling is fulfilled (**B**). Redlund-Johnell: describes the distance between the center of the lower cover plate C2 and the McGregor line. Normal is 34 mm for men and 29 mm for women (**C**).

**Figure 3 jcm-09-00811-f003:**
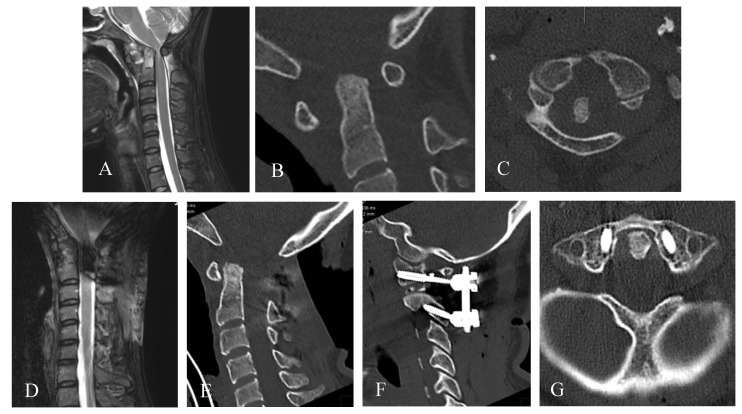
(**A**) Preoperative MRI imaging of the cervical spine atlantodental instability with retrodental pannus formation and basilar impression due to dorsal displacement of the dens resulting in absolute spinal constriction and myelopathy. (**B**,**C**) Enlarged anterior atlantodental interval in the preoperative scan (AADI). (**D**–**G**) Postoperative imaging after initially closed and then open reduction and fixation C1-2.

**Table 1 jcm-09-00811-t001:** Definition and diagnostics of cervical instabilities.

Type of Instability	Definition and Diagnostics of Cervical Instabilities	
	Definition	Diagnostic in Radiograph/Scan in Lateral/Sagittal Projection
AAS (atlantoaxial subluxation)	weakening or rupture of ligaments and subchondral bone erosion in the atlantoaxial joints	anterior atlantodental Interval (AADI) > 3 mm posterior atlantodental interval (PADI) < 14 mm
SAS (subaxial subluxation)	subluxation in the joints C3-7 due to destruction of the joint surface and the ligaments between the processes spinosis	horizontal displacement of vertebrae with irreducible translation > 3.5 mm
CS (Cranial Settling)	vertical translocation of dens into the foramen magnum	see Figure 1 and Figure 2

**Table 2 jcm-09-00811-t002:** Prevalence and symptoms of pat with clinical involvement [6].

Prevalence of Cervical Involvement in RA	%
Pain in the cranio-cervical junction	69% of patients with cervical instability
Muscular atrophy, paresis, bladder rectal disorders, pathological reflexes and spasticity	present in up to 58% of all cases
Involvement of the cranial nerves	reported in about 20%
Initially asymptomatic	33–50%
Atlantoaxial subluxation with myelopathy	circa 2.5% of patients with RA for more than 14 years
Locked-in syndrome or sudden death	rare but reported up to 10% in a postmortem study [13]
Vertebrobasilar insufficiency with tinnitus and dizziness due to Mechanical compression/ vertebrobasilar thromboembolic events due to kinking of vertebral arteries	rare
Aseptic discitis and atraumatic dens fractures	rare

**Table 3 jcm-09-00811-t003:** Ranawat classification [15].

Class	Description
I	Pain, no neurological deficit
II	Subjective weakness, hyperreflexia, dysesthesia
III	Objective weakness, long-tract signs III A—Ambulatory, III B—Non-ambulatory

**Table 4 jcm-09-00811-t004:** Case description.

Case	Case 1	Case 2	Case 3	Case 4	Case 5	Case 6	Case 7	Case 8	Case 9
Sex/age	m/80	w/22	w/82	m/76	m/73	w/71	m/48	w/81	w/50
Cervical pathology	AAI, retrodental pannus	AAS	AAI, retrodental pannus	CS	AAI, retrodental pannus, cervical spinal stenosis subaxial	AAI, retrodental pannus	AAS, retrodental pannus	CS	CS
Pre-operative mJOA - score	9	12	15	8	15	16	14	11	14
Ranawat - criteria	15.5	10.7	12	8.3	12.1	*	13.5	9.5	*
Smallest diameter of canal anterior - posterior (mm)	1.9	4	5.4	5.9	7.2	*	5.1	5.5	*
ADI (mm)	0.2	9.7	0.3	1.3	1.4	*	13.0	0.8	*
Ranawat classification	IIIA	IIIA	IIIA	IIIB	IIIA	II	II	IIIB	IIIA
Rheumatoid arthritis/treatment	suspicion of RA	RA/Adalimumab every 2 weeks	suspicion of RA	RA/MTX every week, folic acid, steroids	suspicion of RA	suspicion of RA	RA (ED 1995)/MTX 1 every week, steroids	RA since 30 years/steroids	suspicion of RA
Symptoms	neck pain, ataxia, hemiparesie, not able to walk	fine motor disorders, monoparesis, bladder emptying disorders, ataxia	monoparesis	tertaparesis, not able to walk, dysphagia, loss of warm cold discrimination of the legs	monoparesie, ataxie	neck pain, ataxie	ataxie, sensory deficit	dysphagia, tetraparesie, not able to walk	ataxie, dysphagia, sensory deficit
Surgery	1. posterior fixation C1-3 + laminectomy C1-2 2. transnasal endoscopic dens resection	closed reduction, osterior fixation C1-2	posterior fixation C1-2, laminectomy C1	posterior fixation C0-2-3-4	posterior fixation C1-2-4-6 + laminectomy C1-6	posterior fixation C1-2	posterior fixation C1-2	posterior fixation C0-3-4, laminectomy C1, decompression suboccipital	1. posterior fixation C0-2, decompression suboccipital 2. transnasal endoscopic resection of dens and clivus (21 months later)
Complications	temporary hemiplegie postoperatively	no	no	deceased	No	no	no	no	no
Follow up	after 8 months, improvement of hemipaesis, walking possible with aide	after 12 months, no symptoms	no follow up	no follow up	no follow up	no follow up	11 months	after 3 months, walking possible	after 5 years
Post-operative mJOA score	12	17	15	deceased	15	16	16	12	16

* Preoperative images not available.

**Table 5 jcm-09-00811-t005:** Demographics.

Number of Patients/Sex	*n* = 9 (Female *n* = 5; Male *n* = 4)
Mean age	64.8 ± 20.5 years (range 22–82 years)
Cervical pathology	AAI with retrodental pannus, *n* = 4 Anterior AAS, *n* = 2 Basilar invagination, *n* = 3 Associated subaxial spinal stenosis *n* = 2
Myelopathy	*n* = 9
Additional neurological deficit	Tetraparesis (*n* = 2) Hemi-or monoparesis (*n* = 4) Incontinence (*n* = 1) Pain (*n* = 4) Dysphagia (*n* = 3)
Pre-operative mJOA- score (mean)	12.67 ± 2.83 (range 8–16)
Ranawat classification	Class II *n* = 2 Class IIIA *n* = 5 Class IIIB *n* = 2
Surgery	posterior fixation, *n* = 9 C1-2, *n* = 4; C1-6, *n* = 1 C1-3, *n* = 1; C0-2, *n* = 1 C0-3, *n* = 1; C0-4, *n* = 1 posterior decompression, *n* = 6 transnasal endoscopic dens resection, *n* = 2
Follow up	mean follow-up at 18.8 ± 23.3 months (range 3–60 months)
Post-operative mJOA score (mean) *	14.6 ± 1.89 (range 12–17)
Mortality	11.1% (*n* = 1)

* evaluated in 5 patients.

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
