# Peer review of "Cervical Myelopathy in Patients Suffering from Rheumatoid Arthritis—A Case Series of 9 Patients and A Review of the Literature"

_jcm, 2020, doi:10.3390/jcm9030811_

Round 1

Reviewer 1 Report

This is an interesting topic regarding cervical spine myelopathy in RA patients who underwent a surgical intervention.

However, there are several aspects that need to be clarified:

The authors present a case series of 9 patients with RA and cervical myelopathy. They should present a short introduction regarding the subject, and then the purpose of their study.

Subsequently, they should describe their methods and the results and finally, the discussion with the relevant literature review. In the present manuscript the authors did the opposite. They have a long introduction with the literature review at the beginning and they conclude the manuscript with a short discussion.

In their methods, the authors should present specific methodological issues. For example: during the period April 2007 - May 2017 – how many patients have been followed and assessed? From this cohort of RA patients how many had cervical spine manifestations, and, finally, how many of them had cervical myelopathy?

 RA has specific classification criteria which help physicians to reach in a definite diagnosis. The authors reported that they evaluated 9 patients with RA or suspected RA… What does this mean? How many of them had definite RA, and how many had other diseases (ie. Ankylosing spondylitis, psoriatic arthritis, or even osteoarthritis)?

There is a confusion regarding the term of “cervical spine instability”, mostly, atlantoaxial instability. AAI may present as atlanto-axial subluxation which has many aspects (posterior, anterior, vertical etc), or as subaxial subluxation. All these must be clarified in the text.

Also, they must describe the role of conventional radiography as a screening test for cervical spine involvement and the role of MRI.

Finally, the authors should present and count the references as they are presented in the text and not alphabetically.

Reviewer 2 Report

The authors summarized and described clearly the cervical involvement in rheumatoid arthritis, making it easy to be understood by a wide audience of medical and healthcare professionals. 

I believe that the time gap between the diagnosis of RA and the onset of symptoms suggestive of cervical involvement should be included in the cases description. Indeed, it would help put the cases reported in the context of the literature reviewed in the introduction. 

On a minor note, I suggest adding a list of abbreviations because it could be helpful to the reader. 

Round 2

Reviewer 1 Report

I have no further comments